# Migrating Lung Monocytes Internalize and Inhibit Growth of *Aspergillus fumigatus* Conidia

**DOI:** 10.3390/pathogens9120983

**Published:** 2020-11-24

**Authors:** Natalia Schiefermeier-Mach, Thomas Haller, Stephan Geley, Susanne Perkhofer

**Affiliations:** 1Health University of Applied Sciences Tyrol/FH Gesundheit Tirol, Innrain 98, 6020 Innsbruck, Austria; susanne.perkhofer@fhg-tirol.ac.at; 2Institute of Physiology, Medical University of Innsbruck, Innrain 52, 6020 Innsbruck, Austria; thomas.haller@i-med.ac.at; 3Institute of Pathophysiology, Medical University of Innsbruck, Innrain 52, 6020 Innsbruck, Austria; stephan.geley@i-med.ac.at

**Keywords:** migrating lung monocytes, patrolling monocytes, non-classical monocytes, alveolar cells, fungal conidia, *Aspergillus fumigatus*

## Abstract

Monocytes are important players to combat the ubiquitously present fungus *Aspergillus fumigatus.* Recruitment of monocytes to sites of fungal *A. fumigatus* infection has been shown in vivo. Upon exposure to *A. fumigatus* in vitro, purified murine and human blood monocytes secrete inflammatory cytokines and fungicidal mediators. Mononuclear tissue phagocytes are phenotypically and functionally different from those circulating in the blood and their role in antifungal defenses is much less understood. In this study, we identified a population of migrating *CD43^+^* monocytes in cells isolated from rat distal lungs. These cells are phenotypically different from alveolar macrophages and show distinct locomotory behavior on the surface of primary alveolar cells resembling previously described endothelial patrolling monocytes. Upon challenge, the *CD43^+^* monocytes internalized *A. fumigatus* conidia resulting in inhibition of their germination and hyphal growth. Thus, migrating lung monocytes might play an important role in local defense against pulmonary pathogens.

## 1. Introduction

The large surface of the respiratory tract is constantly exposed to external environmental factors and elaborate cleaning systems operate to control pathogenic hazards ubiquitously present in inhaled air. The lung is populated with an intricate network of immune cells, including mononuclear phagocytes (MNPs) [1,2,3]. Previous studies have reported that MNPs residing in peripheral tissues are phenotypically and functionally different to circulating monocytes. In particular, isolated lung immune cells provided evidence of several types of MNPs: monocytes, dendritic cells (DCs), resident alveolar macrophages (AMs) and interstitial macrophages [1,4,5]. Whereas DCs and macrophages are well described in terms of ontogeny, function, expression of phenotypical markers and migration [6,7,8,9,10], little is known about lung specific monocytes. Recently described “patrolling” behavior of migrating lung monocytes as well as their localization at the interface between the capillaries and the alveoli suggested an immune surveillance function of these cells [11,12].

Monocytes are increasingly recognized as important cells in the defense against the filamentous and ubiquitously present fungus *Aspergillus fumigatus*. Two classes of circulating monocytes are found in peripheral blood and distinct tissues, such as the lung [11,13,14,15]: “classical monocytes” and “non-classical monocytes (*CD14^+/-^(dim)CD16^++^* in human, *Ly6C^lo^CD43^+^CD62L^-^CCR2^-^* in mice, *CD43^++^CD62L^-^CCR2^-^* in rats)” and some studies define another activated phenotype, the “intermediate monocytes” (*CD14^+^CD16^+^* in human, *Ly6C^int^CD43^+^CD62L^-^CCR2^-^* in mice). They may in fact constitute a third subset, and are, so far, found in men and mice, but not in rats [16,17,18,19]. The relative contribution of non-classical monocytes to the initiation of immunity [20,21,22,23] and/or maintenance of tolerance [24,25,26] implies a dual role that is of crucial interest.

Inhaled conidia of *A. fumigatus* may overcome the upper respiratory tract defense mechanisms and reach the pulmonary alveoli, where macrophages and neutrophils are established as the keystones of host defense [27,28]. Alveolar macrophages were shown to digest and kill conidia, while recruited neutrophils attack *Aspergillus* hyphae that occasionally escaped macrophage killing [29,30]. Recruitment of monocytes to sites of fungal infection has also been shown in vivo [31], but the exact role of monocyte subsets in fungal killing remains unclear. A recent report by Espinosa, Jhingra et al. has shown that depletion of classical *CCR2^+^* monocytes and their derivative DCs in knockout mice decreases *A. fumigatus* conidial containment and results in reduction of neutrophil conidiacidal activity [32]. In vitro experiments using purified murine and human blood monocytes suggested that an *A. fumigatus* infection results in secretion of inflammatory cytokines and fungicidal mediators [33,34]. Upon recognition of fungal β-D-glucan residues by Dectin-1, conidia are phagocytosed by monocytes resulting in inhibition of conidia growth. Monocytes were also shown to secrete TNF-α and activate iNOS in response to *A. fumigatus* [32,33] suggesting initiation of inflammatory responses as well as enhanced killing activity.

In this study, we identified non-classical *CD43^+^* patrolling monocytes in a primary cell mix isolated from rat lungs. We further investigated the function and migratory capacity of these cells in an *A. fumigatus* infection-like setting. Our results suggest that monocytes residing in distal lung tissue have the capacity to (i) migrate on the surface of alveolar cells; (ii) internalize *A. fumigatus* conidia; and (iii) inhibit conidial germination and hyphal growth.

## 2. Results

### 2.1. Cells Isolated form Alveoli Contain Non-Classical Migrating Monocytes

We have modified a previously published method [35] for isolation of primary alveolar type II (ATII) cells from rat lungs to also include MNPs present in this tissue (Figure 1a–d). The cellular isolate was cultivated for 48 h on glass coverslips and found to contain all cell types characteristic for the alveolar tissue, including predominantly ATII, but also ATI cells, fibroblasts and MNPs. Isolated lung MNPs contained *CD45^+^* cells as well as a *CD11b^+^* fraction that may include AMs, DCs, and monocytes (Figure 1a). CD43 was previously shown to be a specific marker of non-classical monocytes in rats [36,37]. We observed a small portion of *CD43^+^* rat monocytes (4.3% of 360 isolated cells, Figure 1a, see Materials and Methods). The morphology of *CD43^+^* cells was strikingly different from that of other cells in this alveolar cell preparation. CD43+ cells were characterized to be smaller with an elongated compact nucleus and a more polarized cell shape (Figure 1a).

Next, we performed live cell video microscopy in order to investigate the motility of immune cells on the surface of adherent ATII cells. By analyzing time-lapse video recordings, we have identified two types of cellular motility: slowly moving round cells that resembled AMs (orange arrows in Figure 1b and Appendix A) and relatively small cells that rapidly moved over the alveolar surface. Due to their small cell size and characteristic “patrolling” migratory phenotype, the latter might be *CD43^+^* non-classical monocytes (red arrows in Figure 1b and Appendix A). We quantified all cells in the field of view and additionally tracked and quantified the slow and fast migrating cells. Results from three independent experiments showed a significant difference in migration velocity (2.30 ± 1.01 vs. 8.5 ± 4.34 µm/min, graph in Figure 1c).

Interestingly, when we seeded the alveolar cell isolate at 50–60% density, cells with patrolling behavior migrated readily on the surface of ATII cells, but avoided migration on empty surfaces devoid of cells (unpublished observations).

In order to further characterize the fast migrating cells that we observed during live cell imaging, we seeded alveolar cell isolates on gridded ibidi dishes. At the end of the live imaging sequence, cells were immediately fixed using warm paraformaldehyde without perturbing the culture to enable position identification of those cells. After staining cells with anti-rat-CD43 antibodies and using the grid reference position, we identified the fast migrating cells as *CD43^+^* non-classical monocytes (Figure 1b). Staining of these cells for rat CD4, CD103, and MPO was negative (data not shown). We observed that nearly 50% of “live” detected fast migrating cells were lost during the fixation process, or cells did move so fast, that we could not completely overlay grid position before and after the fixation process. This might indicate transient or week adhesion of *CD43^+^* monocytes to alveolar cells, which might be important for fast monocyte movement. This result also explains the difference between the percentage of cells with patrolling behavior in live videos that was up to 9% (mean ± 1.75%, results of five independent experiments) and the smaller percentage of *CD43^+^* cells after fixation (4.2% ± 1.2%, results of three independent experiments).

### 2.2. Migrating Lung Monocytes Internalize A. fumigatus Conidia

To further evaluate the role of migrating lung monocytes in pathogen response we challenged the primary alveolar cell isolate with 10^5^ cfu/mL of swollen conidia of green fluorescent protein (GFP) expressing *A. fumigatus* [38]. Live cell imaging experiments showed that migrating monocytes bound and internalized conidia of *A. fumigatus* and further transported them over the surface of ATII cells (Figure 1d, Appendix A). We neither observed germination nor hyphal growth in live videos performed over 8 h, suggesting that the presence of lung monocytes, alveolar macrophages or/and other MNPs efficiently inhibited *A. fumigatus* germination.

### 2.3. Isolated Patrolling Monocytes Inhibit A. fumigatus Conidia Growth

In order to investigate a role of migrating lung monocytes in fungal defense and to exclude the influence of other immune or ATII cells, we used magnetic cell sorting (MACS). We isolated *CD43^+^* cells using CD43-beads and stained isolated cells with CD43 antibody (Figure 2a). From a total of 1.8 × 10^7^ cells in the alveolar cell isolate, CD43 selection resulted in 5.4 × 10^5^
*CD43^+^* cells (3% of all cells in the mix).

Isolated *CD43^+^* cells were plated on IgG-coated glass cover slips/ibidi dishes, since these cells did not efficiently adhere to uncoated surface or other tested coatings, including laminin, gelatin, fibronectin, poly-L-lysine, and the combination of poly-L-lysine and fibronectin (data not shown). By live cell microscopy we could observe migrating *CD43^+^* cells, but in the absence of ATII cells or/and due to the IgG coating, their velocity was significantly decreased (4.59 ± 2.43 vs. 8.50 ± 4.34 µm/min). The morphology, small cell size and polarized cell shape of *CD43^+^* monocytes was clearly different from other lung immune cells as visualized by actin staining (Figure 2b).

Next, we infected isolated *CD43^+^* cells with 10^5^ cfu/mL of swollen *A. fumigatus*-GFP conidia. Live cell imaging experiments showed that migrating *CD43^+^* monocytes internalized and transported internalized *A. fumigatus* conidia (Appendix A) similarly to experiments in Figure 1 and Appendix A. We also observed non-internalized conidia that initially formed hyphae, elongated and further efficiently grew during live microscopy video (Figure 2c, Appendix A). In comparison to non-internalized conidia, internalized ones did not form hyphae and did not grow during the observation period of up to 8 h (red arrows in Figure 2b, Appendix A).

## 3. Discussion

Migrating monocytes were recently discovered as a subset of monocytes that continuously patrol along the walls of blood vessels [17,39]. “Patrolling” locomotion was first described in non-classical monocytes in the microvasculature of the dermis in mice. It was shown to be independent of the direction of the blood flow, and required integrin LFA-1 interactions with ICAM1 [39]. Carlin et al. proposed the enrichment of non-classical monocytes within capillaries and further suggested that these cells may crawl along the endothelial surfaces, phagocytose injured endothelial cells, remove debris, and finally recruit neutrophils to the site of injury [25]. Other studies characterized patrolling capacity with fast locomotion and transmigration into neighboring tissue, suggesting that non-classical monocytes are not restricted to the vasculature [24,39]. In our study, we identified non-classical monocytes in cells derived from the distal lung (“primary alveolar cells”) and show for the first time, that this cellular isolate contains cells that migrate fast on the surface of alveolar ATII cells, crawling, on average, with a speed of 8.5 µm/min under cell culture conditions. One interesting feature of the isolated CD43+ cells was their poor adherence properties, which might be one reason, why they have not been already discovered earlier. These cells adhered only weakly to IgG-coated surfaces but very weakly to other coatings or uncoated surfaces. This poor adherence might however be a prerequisite for their fast locomotion on the surface of alveolar cells as strong adhesiveness would interfere with speed. This phenomenon might rely on certain integrin expression patterns of patrolling monocyte migrating in alveoli. Our preliminary data showed expression of Itgal (integrin alpha L chain), Itgam (CD11b), Itgb1 (CD29) and Itgb2 (CD18) by CD43+ monocytes, with Itgb2 showing the highest expression levels (unpublished observation). The migratory behavior of these cells is also reminiscent of amoeboidal three-dimensional movement as compared to two dimensional movement dependent of focal adhesion complexes. Whether patrolling cell movement is of the amoeboid type and whether any of the mentioned integrins are involved in these processes remains to be investigated.

Monocytes are leucocytes and upon infection, their cell number massively increases. After leaving the circulation and infiltration of the damaged tissue they contribute to the inflammatory response without loss of their monocytic character [11,40]. Tissue monocytes were shown to function in homeostatic tissue surveillance by capturing and transporting antigen to lymphoid organs [11] and serve specific effector functions during infection [34]. A standardized nomenclature and definition of tissue monocytes is complex, since these cells are highly heterogeneous. They can be classified by their function, their gene expression profile or by phenotypical markers. Function, such as combining a critical sensing capacity with a fast locomotion [24,39], is an important factor for the definition of monocytes, irrespective of their progeny. Of note, migrating tissue monocytes, positive for CD43 [41,42], CD14, and/or CD16 expression, can be easily separated from DCs by distinct phenotypical markers, shape and size [11,43,44]. Furthermore, monocytes significantly differ in their more rapid turnover from DCs [45,46].

Tissue monocytes have also been proposed to enter and survey the lung at steady state without differentiation into DCs or macrophages [11]. Murine monocytes have been shown to differ from other MNPs in terms of localization and trafficking: tissue-resident monocytes patrol blood vessels and airways, whereas tissue-resident MNPs solely survey the latter. Regional segregation of monocytes and DCs plays a crucial role, also regarding antigen uptake. This capacity appears to be differently expressed when, in a murine setting, fluorescent beads were applied intravenously or through the airway. In both cases, monocytes were more efficient in bead uptake followed by phagocytosis in comparison to DCs [12]. This observation may point to a specific local environmental interaction of lung capillaries and –alveoli with monocytes. We observed that migrating monocytes are involved in local immune surveillance in alveoli by actively patrolling the alveolar epithelium. Rather than being recruited from the central circulation, we consider the observed cells as “resident tissue monocytes” present in the lung. This assumption is supported by the following pieces of evidence: (a) tissue monocytes can enter and survey the lung without differentiation into DCs or macrophages [11,24,39]; (b) tissue monocytes in lung differ from other MNPs in their localization and trafficking; tissue-resident MNPs only survey the airways, whereas tissue-resident monocytes survey both, blood vessels and airways [12]. To our knowledge, we are the first group that use a primary alveolar cell mix containing alveolar cells and MNPs, as well as the first ones who isolated primary migrating monocytes from the distal lung tissue.

The question concerning the functional differences between blood derived- (recruited) and lung specific (resident) monocytes remains unsolved, particularly for migrating lung monocytes. A significant amount of data was obtained on the process of monocyte transmigration from blood circulation into the lung. In contrast, only a few recent studies pointed to the importance of monocyte localization in lung compartments for immune surveillance and pathogen phagocytosis. Due to the fast locomotion and the ability to internalize pathogens (e.g., fungal conidia), these cells may represent an essential element in the alveolar defense mechanisms against pathogens. Our data suggest that, by infecting the primary alveolar cell mix with fungal conidia of *A. fumigatus,* patrolling cells can internalize multiple living *A. fumigatus* conidia.

In summary, we have observed inhibition of fungal growth inside migrating monocytes. In experiments with MACS-isolated CD43+ cells, we excluded effects of other cell types present in lung tissue such as ATII cells, AMs, DCs, natural killers, and neutrophils, thus eliminating cell-cell interactions and impact of released pro-inflammatory mediators. However, we did not resolve the fate of internalized conidia but using the recently developed fluorescent Aspergillus reporter strain (FLARE) [32] may help to determine conidia cell fate upon internalization into CD43+ cells. Further experiments employing transmission electron microscopy and FLARE strain should then elucidate the fate of conidia within lung migrating monocytes and depletion of CD43+ monocytes in an animal study should clarify the in vivo relevance of patrolling monocytes during fungal infection.

The plasticity of monocytes to respond to pathogenic threats means that different anatomical sites are likely to induce specific phenotypes. The local interaction with tissue specific cells, e.g., lung alveolar cells, as well as the regulation of above mentioned integrin cluster remains enigmatic. Further investigations may re-evaluate the role of tissue specific monocytes in health and disease, and to deepen our knowledge of monocyte patrolling during the course of pulmonary infections.

## 4. Materials and Methods

### 4.1. Reagents

DMEM medium, fetal bovine serum (FBS), L-glutamine, penicillin/streptomycin and phosphate-buffered saline (PBS) were from Capricorn Scientific (Ebsdorfergrund, Germany), EDTA and paraformaldehyde were from Merck (Darmstadt, Germany), IgG-solution, Triton X, Tween®20, DNase, Sabouraud and BSA were from Sigma-Aldrich (Vienna, Austria). Goat serum was from Vector Laboratories (Burlingame, CA, USA).

### 4.2. Isolation of Primary Rat Alveolar Cell Mix

The isolation of alveolar cells was carried out as previously described [35,47] with slight modifications. Briefly, Sprague–Dawley rats were anaesthetized, heparinized, and bled. The lungs were cleared of blood by perfusion and removed from the thorax. After lavage, the lungs were incubated with elastase and trypsin and were finally minced in DNase-solution. After adding FBS to stop the enzymatic reaction, cells were filtered through gaze (2 and 4 layers) and nylon mesh (150 µm, 20 µm and 7 µm) and centrifuged at 130× *g* for 8 min at 4 °C. For live imaging and immunofluorescence experiments, pelleted cells were re-suspended in DMEM, plated onto 10 cm plastic dished coated with IgG-solution and incubated at 37 °C in a CO_2_-incubator for 10 min to remove excessive immune cells. Non-adherent cells were harvested by collecting the supernatant from the petri dishes and washing the cells at 130× *g* for 8 min at 4 °C. The cells were finally re-suspended and cultured in DMEM supplemented with 10% FBS, 2mM L-glutamine, and penicillin/streptomycin at 37 °C with 5% CO_2_. For magnetic cell separation (MACS) experiments, cells were not plated on IgG-coated dishes, thus, increasing the number of MNPs in the cellular isolate.

### 4.3. Magnetic Cell Sorting (MACS)

MACS was performed using MiniMACS separator and MACS columns (Miltenyi Biotec, Bergisch Gladbach, Germany), according to the manufacture instructions. Briefly ATII alveolar cell mix was washed in MACS buffer (PBS, 2mM EDTA, 0.5% BSA), cells were centrifugated at 300× *g* for 10 min at 4 °C, cell pellet was resuspended in 80 µL MACS buffer and 20 µL of rat-specific CD43 MicroBeads, (Miltenyi Biotec, Bergisch Gladbach, Germany), incubated for 15minutes. Further 2 mL of MACS buffer was added following 300× *g* centrifugation at 4 °C for 10 min. Cell pellet was resuspended in 500 µL of MACS buffer, *CD43^+^* cells were separated using columns on ice, washed 3 times in MACS buffer and resuspended in DMEM. Cells were seeded on IgG-coated (IgG resolved in Tris-HCl, pH 9.4, incubated for 4 h at room temperature) ibidi dishes (y-Dish 35mm, ibidi, Munich, Germany) or on glass cover slips, and maintained in growth DMEM. For infection with *A. fumigatus,* the media was replaced with antibiotic free DMEM.

### 4.4. Cultivation and Growth of A. fumigatus

The *A. fumigatus* strain expressing green fluorescent protein (GFP) [38] was grown on Sabouraud 4% glucose agar (15 g/L agar, 40 g/L D(+)-glucose, 10 g/L peptone) plates at 37 °C for 7 days until fully maturation of spores was observed. Conidia were harvested by rinsing the culture surface with sterile distil water containing 0.01% Tween®20. Conidial suspension was washed once in PBS, twice in DMEM without antibiotics, and counted using a hemocytometer. Freshly harvested conidia (10^6^ cfu/mL) were kept in DWEM in shaking incubator at 37 °C, 160 rpm for 2 h to obtain swollen conidia and used in all experiments. For experiments, conidia at concentration 10^5^ cfu/mL were used.

### 4.5. Immunofluorescence

Cells were fixed in 4% paraformaldehyde for 20 min at room temperature and washed three times in PBS. Cells were permeabilized with 0.05% TritonX in PBS for 10 min followed by washing in PBS. Blocking was performed in 10% BSA and 5% goat serum for 30 min. First and secondary antibodies were diluted in PBS containing 1% BSA and 5% goat serum. First antibodies were incubated for 1 h followed by washing in PBS. The following antibodies were used: mouse anti-rat CD43 antibody (Bio-Rad Laboratories, Vienna, Austria), rabbit polyclonal Surfactant C (Proteintech, Manchester, UK), mouse anti-rat CD11b (Bio-Rad Laboratories, Vienna, Austria), mouse anti-rat CD45 (Bio-Rad Laboratories, Vienna, Austria). Secondary antibodies were incubated for 1 h followed by washing in PBS. The following secondary antibodies were used (all from Thermo Fischer Scientific, Vienna, Austria): goat anti-mouse Alexa-FluorPlus488, goat anti-rabbit Alexa-FluorPlus488, goat anti-mouse Alexa-FluorPlus555. DAPI (Sigma-Aldrich, Vienna, Austria) was used for nuclei staining. Pictures were taken using an Oxion Inverso Microscope equipped with precooled CCD camera (both from Euromex, Microscopen, BD Arnhem, The Netherlands) and the ImageFocus 4, v. 2.8 software. Alternatively, images were taken using inverse Zeiss Axiovert 200M (Carl Zeiss, Jena, Germany) equipped with a CCD camera (CoolSNAP HQ2; Photometrics, Tucson, AZ, USA), and acquisition was controlled by the AxioVision release 4.5 SP1 software (Carl Zeiss, Jena, Germany). Live video microscopy was performed using the same Zeiss setup at 37 °C and 5% CO_2_ conditions. Videos and cell tracking were prepared using Fiji free software [48], “Manual Tracking” plug-in (developed at Institute Curie by F. Cordeli for ImageJ Orsay, France) and “Chemotaxis and Migration Tool” plug-in (ibidi, Munich, Germany). Figures were prepared in Fiji [48], and Adobe Photoshop (Adobe Systems Incorporated, San Jose, CA, USA).

### 4.6. Statistics

For quantification of migration velocity, values were transferred from Fiji into Excel program (Microsoft, Redmond, WA, USA) and analyzed by Student’s t-test and one-way ANOVA. Results are expressed as means ± standard deviation (SD) in absolute numbers (µm/min), *p* ≤ 0.001 was considered significant.

### 4.7. Ethical Issues

Alveolar Type II cells were prepared from anesthetized and analogized Sprague–Dawley rats in the laboratory of Thomas Haller. All animal experiments and all steps of cell preparation were approved by the Institutional Animal Care and Use Committee at Medical University of Innsbruck and were conducted in conformity with the Austrian rules for animal care and testing (a license from the Austrian Government, appl. no. BMWFW-66.011/0160-WF/V/3b/2016).

## Figures and Tables

**Figure 1 pathogens-09-00983-f001:**
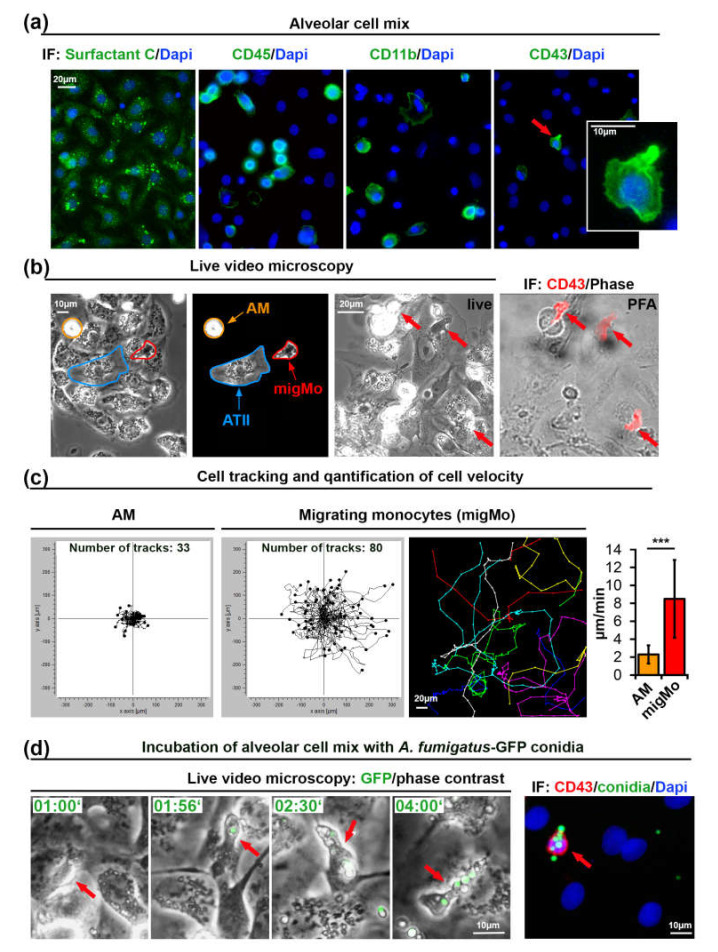
Monocytes isolated from rat alveoli migrate on top of ATII cells and internalize *A. fumigatus* conidia. (**a**) Primary ATII cells express Surfactant Protein C. The alveolar cell mix also contains *CD45^+^, CD11b^+^,* and *CD43^+^* cells. The cell indicated with an red arrow is shown enlarged in the insert; (**b**) Phase contrast live video microscopy revealed round slowly migrating cells (AM, orange markers) and fast migrating monocytes (migMo, red markers). Non-migrating ATII cells are marked in blue. The “live”-labeled image was taken during fixation immediately after the end of live cell video recording with red arrows indicating fast migrating cells with patrolling behavior. The “PFA”-labeled image shows an overlay of a phase contrast image with a CD43 antibody immunostaining in red; (**c**) migratory tracks of AM and migMo measured over 6 h, plotted in XY coordinates. The colored image shows an example of individual tracks in separate colors. The graph depicts the quantification of migration velocity of AM (*n* = 33) and MigMo (*n* = 80). Data are shown as mean ± standard deviation, *** *p* ≤ 0.001; (**d**) migMo (phase contrast, red arrows) bind and internalize multiple *A. fumigatus-*green fluorescent protein (GFP) conidia (in green). Time is indicated as “hours:minutes”. The most right image depicts CD43 staining (red), GFP-conidia (green) and DAPI (blue). IF—immunofluorescence, PFA—Paraformaldehyde, AM—alveolar macrophages, ATII—alveolar type II cells, migMo—migrating monocytes. See also Appendix A.

**Figure 2 pathogens-09-00983-f002:**
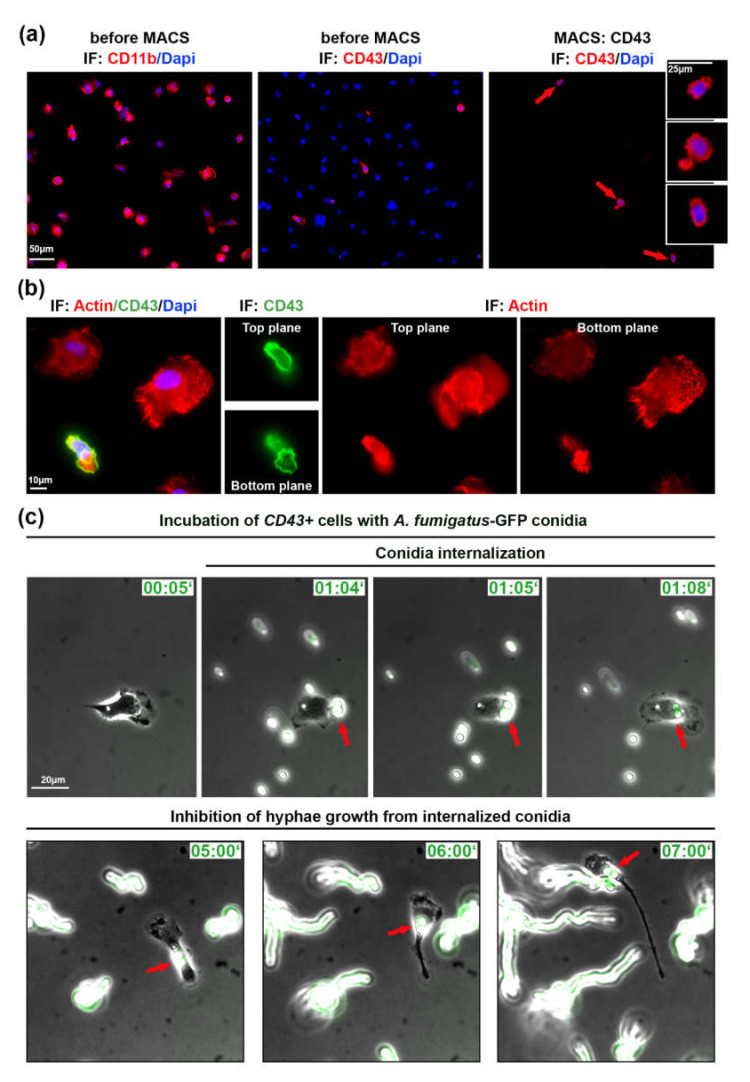
Isolated patrolling monocytes inhibit germination of internalized *A. fumigatus* conidia. (**a**) Cells before and after magnetic cell sorting (MACS). Cells indicated with red arrows are depicted in inserts. IF: CD43, CD11b in red, DAPI in blue; (**b**) cell morphology of immune cells plated on IgG before MACS. Two focal planes are depicted. IF: phalloidin staining in red, CD43 in green, DAPI in blue; (**c**) Isolated *CD43^+^* cells infected with swollen *A. fumigatus-*GFP conidia (green). Shown are phase contrast images merged with GFP channel. Red arrows point on conidia binding and internalization into *CD43^+^* cell. IF—immunofluorescence, Time is indicated as “hours:minutes”. See also Appendix A.

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
