# Peer review of "Migrating Lung Monocytes Internalize and Inhibit Growth of *Aspergillus fumigatus* Conidia"

_pathogens, 2020, doi:10.3390/pathogens9120983_

Round 1

Reviewer 1 Report

            The article “Migrating lung monocytes internalize and inhibit growth of Aspergillus fumigatus conidia” presented by Natalia Schiefermeier-Mach, Thomas Haller, Stephan Geley, Susanne Perkhofer, is devoted to the study of role of migrating monocytes in innate immunity.  Authors studied a population of migrating CD43+monocytes in cells isolated from rat distal lungs. They studied locomotory behavior of migrating CD43+monocytes on the surface of primary alveolar cells (patrolling activity) and the interaction of CD43+monocytes  with conidia of Aspergillus fumigatus.

            Isolation and study of the behavior of migrating monocytes in a mixture of living cells, as well as the description of their interaction with living conidia, is a difficult experimental task. The authors of the article coped with this task and obtained interesting results. They identified monocytes using immunofluorescence, measured the rate of movement of monocytes in cell culture and recorded interactions with conidia using live video.  

Comments and suggestions  

  1. The article did not demonstrate that the conidia were internalized (ingested). Conidia can attach to the cell surface and the microscopic image will be the same. Transmission electron microscopy, but not fluorescence microscopy, can distinguish between internalization and attachment. Authors should have used a different term — binding, scavenge, or otherwise.

  1. Figure 1 a. The image of CD43 / Dapi cells is inconclusive. The cells are too small and appear to be destroyed, since CD43, marked in green, and Dapi (nucleus), marked in blue, are practically separated. It is worth picking up more successfully stained cells. Since the cells are small, it is worth doing a high magnification insert.

  1. Figure 2 a. Images of CD11b / Dapi and CD43 / Dapi cells before magnetic sorting of cells (left and middle photographs) completely repeat the images of the same cells in Figure 1a (two right photographs). Why duplicate pictures in a small article?

  1. The authors claim to have isolated a pure population of CD43 cells using magnetic sorting. “CD43 selection resulted in 5.4x105 CD43+ cells (Line 128)”. It would be convincing to present a photograph showing at least a few CD43+ cells of an isolated population in the field of view. The photograph provided by the authors (Figure 2 a, photo on the right) shows a single cell, and the image of CD43 / Dapi cell after MACS is also inconclusive. The CD43 cell appears to be aggregated with another small-nucleus cell not labeled with anti-CD43 antibodies.

  1. Authors wrote that migrating monocytes internalized and inhibit growth of Aspergillus fumigatus conidia based on the experiment with the single cell. The conidia attached or ingested by this cell may be dead initially. It is necessary to observe the rate of conidia in the number of the CD43 cells to make the significant conclusion.

  1. Lines 12: “monocytes are able to induce inflammatory and fungicidal mediators as 12 well as the host cell and the fungal transcriptional responses”…

Line 55: “A. fumigatus infection results in induction of inflammatory and fungicidal mediators”…It is not clear what it means? Secretion or expression of inflammatory mediators can be induced.

Reviewer 2 Report

In this communication, Schiefermeier-Mach and colleagues present interesting data on an important topic, i.e., the identification and initial characterization of migrating lung monocytes during infection with conidia of the pathogenic fungus Aspergillus fumigatus.

CD43+ monocytes were isolated from rat lungs. These cells showed the capability to migrate during confrontation with fungal conidia. Migrating monocytes were able to phagocytose conidia and to inhibit germination and thereby hyphal growth.

Major comments:

It remains unclear, whether migMo cells inhibit germination of conidia or kill internalized conidia. Also the authors did not observe lysis of conidia, this does not necessarily indicate that the conidia are still alive. Also in this regard, it was not studied, whether after phagocytosis of conidia the phagosomes maturate and acidify to become fungicidal phagolysosomes. Such studies on the fate of conidia after phagocytosis, e.g. by applying the A. fumigatus FLARE strain, would clearly strengthen the manuscript and support the hypothesis that migMos play an important role in antifungal defence.

The authors should at least discuss the possibility to deplete CD43+ monocytes (blocking by antibody) in an animal infection study to analyze their in vivo relevance during infection. 

Minor comments:

Formatting of "A.fumigatus" has to be corrected throughout the manuscript to "A. fumigatus" (italics and space between "A." and "fumigatus").

Change "iBidi" to  "ibidi" and "Dapi" to "DAPI".

Reviewer 3 Report

This manuscript shows that the CD43 positive monocytes internalized live A.fumigatus conidia resulting in inhibition of conidial germination and hyphal growth. This study provides meaningful information that shows that the migrating CD43+ monocytes are phenotypically different from alveolar macrophages. By using live video microscopy, the authors showed a significant difference in migration velocity between the round cells and small cells, which may be alveolar macrophages and CD43+ non-classical monocytes, respectively. The experimental details are well addressed and the logic of the order of presentation is entirely clear. Additionally, previous studies have shown the CD43+ lung macrophages belong to infiltrating immature M1 monocytes/macrophages, and CD43 is required for growth inhibition of mycobacterium tuberculosis in macrophages. These conclusions are consistent with the data presented in this study.

One concern is from the data shown in Figure 1b. To clearly distinguish the ATII cells, alveolar macrophages, and CD43+ monocytes, it would be better to label the cells with specific markers in independent experiments.

Round 2

Reviewer 2 Report

The authors addressed my points sufficiently.